# Synchronizing the Logic of Inquiry with the Logic of Action: The Case of Urban Climate Policy

**Chris J. Barton** [1], **Qingqing Wang** [2,3,*], **Derrick M. Anderson** [2] and **Drew A. Callow** [4]

1    School for the Future of Innovation in Society, Arizona State University, Tucson, AZ 85721, USA; cjbarton@tbird.asu.edu
2    School of Public Affairs, Arizona State University, Tempe, AZ 85281, USA; Derrick.Anderson@asu.edu
3    The Hainan University-Arizona State University Joint International Tourism College, Hainan University, 58 Renmin Road, Haikou 570004, China
4    Greater Phoenix Economic Council, Phoenix, AZ 85004, USA; dcallow@gpec.org
*    Correspondence: Qingqing.Wang.1@asu.edu

**Abstract:** Policymakers often rely on scientific knowledge for making policy decisions, and many scientists aim to produce knowledge that is useful to policymakers. However, the logic of action (which guides policy) and the logic of inquiry (which guides research) do not always align. We introduce the term "logic synchronization" to characterize the degree to which the logic of policy action aligns with the logic of scientific inquiry. We use the case of urban climate policy to explore this dynamic using a purposive literature review. The framework presented here is helpful in identifying areas in which the logic of inquiry and the logic of action synchronize, creating the opportunity for both policy-relevant science and science-informed policy. It also reveals where the logics do not yet synchronize, which indicates where scientists and policy makers can productively focus their efforts. The framework introduced here can be both theoretically and practically useful for linking scientific knowledge to policy action.

**Keywords:** public policy; climate change; urban climate policy; policy-relevant science; science-informed policy





## 1. Introduction

It is not a given that scientific research about a policy relevant topic will actually be included in the policy process. It has been observed across many policy domains that the knowledge which is produced by scientists is not always the knowledge which is needed by policymakers [1–12]. The supply and demand of knowledge are disconnected: increasing the supply of knowledge does not automatically lead to increased demand for that knowledge [7]. In other words, the production of knowledge does not guarantee its use.

While tracking the cause of this disconnect between the creation and use of scientific knowledge, McNie surveyed the literature and concluded that "by concentrating efforts on increasing the supply of scientific information, scientists may not be producing information considered relevant and useful by decision makers [4]." The logic which informs scientists as they produce knowledge is not the same logic which informs decision makers as they decide how to act.

This paper introduces a new approach to understanding this persistent disconnect between the production and use of knowledge, and a novel framework for exploring and navigating it in the context of specific policy domains. We propose that the synchronization of the 'logics' which guide scientists and policymakers is a pre-requisite for the utilization of science in policy formation.

We see this work as extending and supporting the burgeoning literature on strategies for connecting knowledge to action, specifically that which deals with co-production of knowledge and the production of actionable science [13–20]. Actionable science is that

which is designed for, and capable of, influencing decision-making [13,14,21]. Actionable science is understood to inform action rather than prescribe action—it is understood as one of many sources of information used by decision makers [14]. Recent years have seen much scholarship on how scientists can go about creating actionable science, with co-production—the process of producing knowledge in collaboration with stakeholders, including policymakers—arising as a promising model [14–17,21].

However, this literature does not address the problem that making science *actionable* does not necessarily mean that it will be *acted upon*. Even if effective co-production can produce actionable science, there is no guarantee that the science will be taken up and used by practitioners. Action requires neither complete nor accurate information. The epistemological question "what do we know?" and the ethical and practical question "what should we do?" are separate questions, answered in different ways. The former animates *the logic of inquiry*, and the latter animates *the logic of action*.

Public policy is concerned primarily with the second question. Scientific evidence is only one source of information used to answer this question, alongside "personal and political beliefs and values, . . . lessons from experience, trial and error learning, and reasoning by analogy" [22]. Decades ago, Carol Weiss observed that sometimes knowledge produced by scientists "[bounces] off the policy process without making much of a dent" [9]; today a similar frustration has led to interest in concepts such as actionable science. Yet if the logics of inquiry and action are not aligned, science may remain excluded from policymaking regardless of how actionable it is.

Urban climate policy offers a promising case study with which to demonstrate how the logics of inquiry and action can align, a process we are calling *logic synchronization*. This policy domain is relatively well-defined, both scientists and policymakers generally agree on its importance, and there is a long-standing intention to create both policy-relevant science and science-informed policy. These features make this a domain which benefits from relatively high levels of synchronization. But even so, recent literature in urban climate science has again highlighted the rift between science and policy [8,23–26]. The persistent disconnect between scientists and policymakers makes urban climate policy a useful case study.

Following a theoretical introduction to the logics of inquiry and action, this paper presents a purposive literature review as the basis for an illustrative assessment of logic synchronization in the domain urban climate policy. We then further discus the merits of this approach, its limitations, and possibilities for future research.

*The Logic of Inquiry versus the Logic of Action*

We propose that the disconnect between science and policy can be partially explained as a disconnect between the logic of inquiry and the logic of action. Others have observed that scientists can be thought of as belonging to a community concerned primarily with "pure science and esoteric issues," while "government policy makers are action-oriented, practical persons concerned with obvious and immediate issues" [2]. This so-called "two communities" perspective provides a useful lens for understanding the problem, but only a partial explanation. We expand on this perspective with the observation that complex policy systems involving multiple groups of actors also include multiple logics. As a matter of terminology, we propose that scientific knowledge is generated according to *the logic of inquiry* while policy decisions are made according to *the logic of action*.

In the context of policy systems, we can define "logics" as systems of principles, reasons, and arrangements which guide action. Speaking of the role that logics play in the design of universities, Crow and Anderson purport that logics answer questions like: What is our purpose? How do we accomplish it? What comes next? [27] Logics exist at the scale of the individual as well as that of the organization and the community.

*The logic of inquiry*, which generally guides scientific communities, can therefore be defined as the system of principles, reasons, and contextual arrangements that determine

what questions are asked, what methods are employed to answer those questions, and how the results are presented and disseminated.

Similarly, *the logic of action* generally guides communities of policymakers, and can be defined as the system of principles, reasons, and contextual arrangements that determine which problems require attention, the methods of determining what actions should be taken in response to those problems, and the enactment of those actions within a given context.

The difference between these logics is complementary to the distinction Sheila Jasanoff finds between 'regulatory science' and 'research science,' where the former aims to inform the policy-making process while the latter aspires to uncover significant and original 'truths' [28]. Jasanoff's focus is on exploring the controversies that arise around scientific issues in the regulatory process; our logics framework moves beyond controversies to also recognize missed opportunities and unsuccessful collaborations. Regulatory and research science are recognized where they intersect in adversarial ways, while the logics of inquiry and action often fail to intersect at all. Where Jasanoff's framework explains antagonism and controversy, ours explains silence and ignorance. We believe that our logics framework complements Jasanoff's distinction between regulatory and research science, as the logics framework applies to policy actors beyond those who identify as doing science (whether of the regulatory or research variety), and it allows us recognize areas where interaction does *not* happen.

The logic synchronization framework can be useful for understanding the dynamics of many domains with many types of actors. Although the logic of inquiry tends to guide scientists and the logic of action tends to guide policymakers, these logics can inform the actions of any group. For example, think tanks and research institutes (such as the Brookings Institute or the Pew Research Center) are likely guided by the logic of inquiry while many NGOs with concrete operations (such as the Red Cross or the International Rescue Committee) follow the logic of action. Both logics can even manifest in the same organization: a corporate research and development department might be guided by the logic of inquiry whereas the rest of the organization adheres to the logic of action. Sometimes similar groups can adhere to different logics: the example given below, of scientists working on the Deepwater Horizon oil spill, demonstrates this.

In each of these situations there are likely to be areas of disconnect, where the two logics are out of sync and the relevant stakeholders are not effectively working together. These areas are not always apparent, since each group, guided by their own logic, may have a hard time recognizing what the other group needs and values. Our framework offers a process by which these blind spots and disconnects can be identified and addressed.

An example may help clarify these ideas. In his work on the containment and clean-up of the 2010 Deepwater Horizon oil spill, David Bond identifies two groups of marine scientists involved in surveying the disaster. Although these two groups had ostensibly similar training and background, they self-identified into two distinct communities: "academic scientists often described themselves as belonging to a 'research community' while government scientists emphasized their commitment to solving real problems with science" [29]. The academic scientists (Jasanoff's research scientists) were operating under the logic of inquiry, while the government scientists (Jasanoff's regulatory scientists) adhered to the logic of action.

The two groups, having access to the same empirical evidence, initially disagreed about the extent of the oil spill and what parts of the environment were most at risk. Bond identifies this disagreement as a result of how each group produced new knowledge. "Government scientists . . . sought to produce operational facts that could help rein in an unfolding event," while the academics "sought to produce disciplinary facts, focusing attention on documenting the specific impact of dispersed hydrocarbons on the ecological niche or species they knew best" [29]. While the government scientists were focusing on containing the oil that rose to the surface, academic research on 'deepwater plumes' revealed the myriad ways that hydrocarbons in the deep ocean were changing the chemical and biological conditions underwater. Academic scientists found compelling reasons to

look at the oil spill as a three-dimensional disaster, whereas government scientists initially limited their scope to the two-dimensional ocean surface.

For several months "the mandate to produce useful knowledge led government scientists (and federal officials) to actively exclude microbial and chemical evidence of deepwater plumes" [29]. The science being done on deepwater plumes was of immediate relevance to those dealing with the disaster, and the academic scientists were in direct communication with the response team. Yet the logic that called for action to stem the disaster was independent from the logic that called for research to understand it. The logic of action and the logic of inquiry were asynchronous, resulting in the exclusion of knowledge about deepwater plumes from the decision-making process. The issue here was not one of contested facts; no one was arguing that deepwater plumes did not exist. Rather, the issue was that research which seemed important under the logic of inquiry was considered unimportant under the logic of action. The former situation is well-described by Jasanoff's model, while we believe the latter is better described with ours.

We believe a similar dynamic is at work in many policy areas. Where the logics of action and inquiry are synchronized, policy-relevant science is produced and is used by policymakers. Where the logics are divergent, scientists and policymakers find themselves working in separate worlds and talking past one another. This cannot be attributed entirely to ineffective communication techniques or organizational barriers to collaboration (though these do exist); at issue is a fundamental difference in understanding what work is important, why it is important, and how to go about doing it.

## 2. An approach to Logic Synchronization

In order to explore these concepts further, this paper takes urban climate policy as a case study and presents a preliminary map where the two logics seem to converge, leading to the production of policy-relevant scientific knowledge. Our objective in this paper is not to explain why or how the two logics synchronized in this case—that would require an understanding of the dynamics of urban climate policy beyond our limited expertise—but rather, we aim to demonstrate how our framework can shed light on *where* the logics do and do not synchronize. The insights revealed by this analysis are discussed later in the paper.

The first step in examining logic synchronization is defining the scope of policy-relevant research and the scope of possible policy action. For this paper, policy-relevant research relating to the effects of climate change on cities was collected and organized according to dominant themes. We reviewed both peer-revied academic literature (associated with the logic of inquiry) and grey literature such as municipal reports and white papers (associated with the logic of action).

The literature used for this exercise was collected primarily via an online search, using Google, Google Scholar, and the ASU library search tool (more info on which databases this includes can be found here: https://libguides.asu.edu/primo-info). Both peer-reviewed academic articles and policy documents were sought out, hence the use of both Google and Google scholar; the ASU library returned both kinds of documents.

Search results were prioritized based on their impact (as determined by number of citations or rank in the google search) and their applicability to a broad understanding of the effects of climate change globally. We collected papers until we felt we had reached either theoretical saturation in a particular area or when all high-impact papers had been identified.

The main search terms used were "Climate Change," "City," and "Cities" along with "Public health," "Human Health," "Biodiversity," "The Environment," "Government," "Public policy," "Public Management," "Education," "Quality of Life," "Infrastructure," "Water resources," "Economics," "Economic Development," "Urban Development," "Energy," "Natural Disasters," "Coastal Populations," "Agriculture," "Farming," and "Technology".

This review was conducted in 2017, with the earliest notable literature coming from the early 1990s. There was not a specific focus on geographic area, since the effects of

climate change are global in nature; however, the English-language literature did tend to focus on the US and other developed, democratic countries.

To establish the scope of policy action, we reviewed the structure of the 10 largest cities in the US (New York, Los Angeles, Chicago, Houston, Phoenix, Philadelphia, San Antonio, San Diego, Dallas, and San Jose), as well as the 10 largest cities in China (Shanghai, Beijing, Tianjin, Shenzhen, Guangzhou, Chengdu, Chongqing, Dongguan, Shenyang, Wuhan). The US and China were chosen because they are the world's largest and most influential developed and developing countries. Based on their organizational structure (i.e., named municipal departments), we identified ten broad areas of action. The resulting domains of research and action are listed below. These lists are not intended to be comprehensive. Since our primary goal is to demonstrate our tool, we limited ourselves to a manageable number of categories. We acknowledge that both climate change research and urban policy are much more complex than we are presenting them here.

We organized the research on climate change into the following general categories:

- Ecosystems and Biodiversity
- Extreme Weather
- Food Security
- Precipitation
- Sea Level and Coastal Areas
- Temperature
- Water Quality and Supply

We organized the policy actions of a city government into the following general categories:

- Community Affairs
- Economic Development
- Education
- Energy
- Emergency Management
- Housing
- Intergovernmental Relations
- Infrastructure and Urban Planning
- Public Health and Safety
- Water and Sewer

The second step is organizing these domains in a grid, with the climate change research categories on one axis and the domains of government action on the other. We then recorded where we were able to locate interaction between these domains in the literature. The interaction map below (Table 1) shows the presence of work done at the intersection of a climate change research category (columns) and an area of local government action (rows). We take the existence of research at the intersection of these domains to indicate instances where the logics of inquiry and action are in sync.

We chose to use a binary scale (synchronized/not synchronized) to represent where we identified logic synchronization, in the form of policy-relevant science. Although not all areas had comparable levels of interaction (which can be seen in the discussion section below), our review of this literature was not systematic and thus we are unable to make claims about the relative volume of work done in each of these areas. A more comprehensive analysis may benefit from a more nuanced scale. Whether this scale represents the quantity or the quality of work done in each area (or some combination of the two) would depend on the needs of the person or group doing the analysis; for example, one might record the number of journal articles in each area, but weight each article based on its relevance to a particular policy domain.

**Table 1.** Areas where the logic of inquiry and the logic of action synchronize in the domain of urban ×.

| | | Logic of Inquiry | | | | | | |
| --- | --- | --- | --- | --- | --- | --- | --- | --- |
| | | Ecosystems and Biodiversity | Extreme Weather | Food Security | Precipitation | Sea Level and Coastal Areas | Temperature | Water Quality and Supply |
| Logic of action | Community affairs | | | | | × | × | |
| | Economic Development | | × | × | | | × | × |
| | Education | | × | | | | × | |
| | Energy | | × | | × | × | × | × |
| | Emergency Management | × | × | × | × | | | |
| | Housing | × | × | × | × | × | × | × |
| | Intergovernmental Relations | | | | | × | | |
| | Infrastructure and Urban Planning | | × | | × | × | × | × |
| | Public Health and Safety | × | × | × | × | | × | × |
| | Water and Sewer | | × | | × | × | | × |

We want to be clear that these insights are based only on preliminary assessment of a purposive, rather than systematic, sample of the literature. Our intention is primarily to demonstrate our tool, and therefore a full review of this literature is beyond the scope of this paper. Urban climate science is a dynamic, interdisciplinary field which brings together climate scientists, social scientists, policy scholars, and urban planners. For an excellent overview of the field and its current state see Rosenzweig and colleagues' work with the Urban Climate Change Research Network [30].

## 3. Making Sense of Interacting Logics

Below we summarize the literature on climate change which relates to each domain of government action. Each section corresponds to a row on the table above. We explore where the logics of inquiry and action seem to be synchronized, as indicated by the availability of literature which speaks to both a domain of action and a domain of inquiry.

### 3.1. Community Affairs

This area of action interacts weakly in the literature with *temperature* and *sea level and coastal areas*. This can be perhaps attributed to the fact that it is difficult to assess the impact of climate change on issues related to community affairs in local government; impacts may be indirect at best.

However, there seems to be increasing interest in community engagement around climate change, as public officials turn to 'participatory governance' or a 'bottom-up' approach in working to solve climate issues. For example, governments can partner with grassroots organizations and work with the public to increase emergency preparedness [31].

Due to climate change, many settlements and arable lands are at increased risk from flooding caused by rising sea level [32]. Some cities may be forced to consider managed retreat as coastlines naturally recede. However, this will present challenges associated with property rights and mass implementation, and will necessarily involve community engagement [33].

### 3.2. Economic Development

Sparse research could be found linking economic development with extreme weather, food security, temperature, and water quality and supply.

An analysis found that the budgetary impact of extreme weather events seems to be more pronounced in developing countries that in advanced economies [34]. In advanced economies, the budgetary impact of extreme weather events seems to have had a limited magnitude in terms of GDP and, therefore, also limited impact on the sustainability of public finances in the long term [34].

So-called 'hardiness zones' used by horticulturists have begun to change because of temperature increases, and the habitable zones of many North American species are moving northward [35]. This effect may have an impact on land allocation for farming and agriculture and increase farmers' reliance on government subsidies, although subsidies are largely at the federal rather than local level.

A limited amount of research evidence indicates that increasing temperature will coincide with an increase in aggregate economic losses. These losses are difficult to estimate [33]. There is little evidence of immediate direct impact of climate change on economic development outside of the occurrence of natural disasters. In a working paper by the Commission on Growth and Development, Mendelsohn argues that the most significant impacts of climate change on economic development in the near future will be excessive near-term efforts to mitigate climate change impact [36]. Evidence from one study indicates that rising electricity costs will indirectly create economic losses as business transaction costs increase and the distribution of resources shifts or becomes unbalanced. The same study argues that rising temperature may result in a reduction in GDP by three to six percentage points [37].

Water quality is increasingly constraining developing countries' agriculture and economic growth through adverse impacts on water availability for urban-industrial growth [38]. Fishing and shellfish industries are hurt by harmful algal blooms that kill fish and contaminate shellfish [39]. Annual losses to these industries from nutrient pollution are estimated to be in the tens of millions of dollars [39].

### 3.3. Education

There are relatively few studies focusing on the effect of climate change on *education* systems. The only areas of interaction are *extreme weather* and *temperature*.

One study found that individuals who were of school age and living in severely affected districts during extreme weather shocks were significantly less likely to have completed mandatory education, compared to peers in less affected districts [40]. Another study conducted on the temperatures in school classrooms suggested that increased temperatures and poor air quality/ventilation cause headaches and poor academic performance [41]. Under this assumption, it is possible that increased temperatures from climate change could lead to poor academic performance of students in hotter school classrooms, or higher energy costs for schools that chose to decrease room temperature.

### 3.4. Energy

*Energy* interacts with climate effects in much the same way as the *infrastructure and urban planning* category, perhaps because studies tend to focus on how climate change will influence energy infrastructure. *Energy* interacts with *extreme weather, precipitation, sea level and coastal areas, temperature,* and *water quality and supply.*

For example, several thousand oil drilling platforms offshore of the Gulf Coast are vulnerable to extreme weather events [42]. A large portion of U.S. energy infrastructure is located in coastal areas and therefore sensitive to sea level rise and storm surge [42]. A substantial portion of U.S. energy facilities are located on the Gulf Coast or offshore in the Gulf of Mexico. Several coastal power plants in the U.S. are less than three feet above sea level, and facilities that import or export coal, gas, and oil are also located in coastal regions. Sea level rise and more intense storms and hurricanes in coastal areas could increase the risk of energy supply disruptions [42].

Climate change can also affect energy demand. One study implies that the negative or positive effects of climate change would be dependent on time of year and income level. The temperature of a given geographical area and the time of year will determine whether energy resources such as electricity, gas, coal, oil products, etc. are in higher or lower demand upon temperature changes [43].

Climate change will also reduce the efficiency of many fossil fuels and nuclear power plants because these facilities need to use water for their cooling systems. Increased temperatures, increased evaporation, and droughts may increase the need for energy-intensive methods of providing potable water for irrigation, such as desalination plants that convert saltwater to freshwater but consume large amounts of energy [44]. Moreover, flooding and intense storms can damage power lines and electricity distribution equipment [42].

### 3.5. Emergency Management

There is significant but focused literature exploring the interactions between city level *emergency management* and *ecosystems and biodiversity, extreme weather, food security,* and *precipitation.*

Higher temperatures could lead to heat waves and wildfires, among other extreme weather events, and these events could cause massive migration within the U.S., as well as conflict over resources and shifting disease patterns [45]. Increases in forest fires will also result in increases in local government expenditures to combat and mitigate wildfire effects and harm to property and the public. Changes in precipitation patterns, including increases in the frequency of sudden, extreme precipitation events, pose some of the most

significant challenges for state agencies and hazard planners, threatening increases in loss of human life, damage to infrastructure, and the spread of disease [46].

Drought also poses risks to public safety. Drought conditions may increase the environmental exposure to a broad set of health hazards including wildfires, dust storms, extreme heat events, flash flooding, degraded water quality, and reduced water quantity [47]. Changes in precipitation and unfamiliar climate patterns severely decrease officials' ability to accurately predict and plan for floods, droughts, and other natural disaster events, so local hazard mitigation and emergency operations plans need to be revised to take increased uncertainty into account [46]. Moreover, governments will likely see an increase in costs associated with crop losses due to changes in weather. Between 1981 and 1994, the Federal Crop Insurance Corporation paid roughly $5.2 billion in claims excess of producer premium. That amount increased to $14.2 billion from 1994 to 2005 [48].

### 3.6. Housing

We found literature integrating *housing*-related actions with all seven categories of climate-change effects. Notably, studies have shown that the effect of climate change on *ecosystems*, *extreme weather*, *food security*, and *water supply* could eventually lead to price fluctuations in the housing market.

Urban sprawl will be increasingly problematic as it compounds the effects of climate change on ecosystems and reductions in biodiversity [49]. Modernized urban centers could increase vertical rather than horizontal housing growth, increasing population density and reducing the negative impacts of sprawl on native wildlands. The real estate industry is inevitably exposed to the potential loss of property returns in locations that are vulnerable to extreme weather [50]. Housing development patterns could change to reduce the Wildland-Urban Interface (WUI) which is where the most structures are destroyed by wildfires [49].

In a study researching the relationship between food security and housing security, 42.4% of those with food insecurity also reported housing instability [51]. The effects of increased food insecurity may therefore be compounded by the effects of climate change on housing.

Other studies have shown that flood, droughts and water scarcity affect the housing market. The housing sector might suffer from uninsured flood damage to buildings in the short run, but reconstruction can also have positive effects in the longer run [52]. Droughts are assumed to have only minor effects on the housing sector [52]. Rising sea levels can reduce the settlement area in coastal regions [50] leading to significant losses in the housing market. Water scarcity can cause a decline in attractiveness of a region, decrease rent values, and increase costs for water supply and treatment [50]. Changes in regional desirability will subsequently impact tax revenues, forcing local governments to adapt.

### 3.7. Intergovernmental relations

This domain of action interacts in the literature only with *sea level and coastal areas*. This may be because cities in coastal regions have a strong need to cooperate with other governments in order to deal with rising sea levels. New York City is an example of this type of cooperation, as the city's Department of Environmental Protection Climate Change Task Force has worked to bring in groups and agencies from other jurisdictions within the coastal region to discuss shared infrastructure and coordinated operations [53].

### 3.8. Infrastructure and Urban Planning

Like energy, this category interacts with extreme weather, precipitation, sea level and coastal areas, temperature, and water quality and supply.

Changes in the frequency and severity of storms and other extreme events may damage energy infrastructure, resulting in energy shortages that harm the economy and disrupt people's daily lives [42]. The total global damage caused by extreme weather, mainly to property and infrastructure, is rising dramatically and now exceeds $150 billion

per year [50]. Predicted increases in the frequency and severity of flood events are likely to compromise human infrastructure located on floodplains or along rivers and streams. This includes roads, bridges, and buildings which may be inundated or damaged by fluvial erosion or landslides [46]. Water intrusion into buildings can result in mold contamination that manifests later on, leading to indoor air quality problems [47]. Buildings damaged during hurricanes are especially susceptible to water intrusion [47]. Rising sea levels can create danger to port facilities [50]. Water scarcity can cause a decline in the bearing capacity of soil, creating risk for infrastructure [50]. Transportation and energy infrastructure will likely be burdened with greater energy demands and higher damage risks due to increased temperature variability [54]. Research calls for cities to conduct urban design initiatives to develop infrastructure that more effectively mitigates heat retention and urban heat island effects [55].

### 3.9. Public Health and Safety

This category of action has high levels of interaction in the literature with each climate change research area, with the unexpected exception *of sea level and coastal areas*. It is here that we see the strongest alignment between the logic of action and the logic of inquiry.

Expansion of tropic and sub-tropic climates towards the poles could potentially invite deadly diseases like malaria and dengue fever into North America [56]. Pathogen exposure is likely to be amplified by increased population concentration within city centers. Changes in precipitation may increase the spread of vector-borne disease and create climate-related epidemics [32]. Increasing threats to human health from disease outbreak will require local governments to improve timeliness and comprehensiveness of public healthcare and especially health surveillance systems [55]. Increased precipitation and flooding will increase the spread of pest species and disease. Expanding urban areas will likely see increased health challenges as sewer systems and wastewater treatment infrastructure is inundated by increased frequency of heavy precipitation, leading to increases in illness vectors, toxic compounds and pollutants [57,58]. Change in precipitation may also affect the frequency and severity of cyanobacteria blooms, as well as the spread of insect-borne diseases such as Lyme disease and the West Nile Virus [46].

Heat waves, floods, and droughts are all capable of directly impacting human welfare and mortality [59]. Heat is currently the leading cause of weather-related fatalities in the US [60]. The risk of heat exhaustion and heat stroke will increase in arid regions as global temperatures increase, particularly in the southwest [55]. A large portion of excess mortality from heat is related to complications with cardiovascular, cerebrovascular, and respiratory disease and is concentrated in the elderly and individuals with preexisting conditions [59]. Therefore, evidence shows that changes in public health preparedness measures can reduce heat wave morbidity and mortality rates to some degree [32].

Floods are the second deadliest of all weather-related hazards in the United States, accounting for approximately 98 deaths per year, most of them due to drowning [47]. Populations living in damp indoor environments from flooding experience increased prevalence of asthma and other upper respiratory tract symptoms, such as coughing and wheezing, as well as lower respiratory tract infections, such as pneumonia, respiratory syncytial virus (RSV), and RSV pneumonia [47].

Food and water security also have huge impacts on public health and safety. Food insecurity is associated with postponing necessary health care and medications [51]. For example, diabetics with hunger have increased risk of hypoglycemic episodes and increased health care utilization [51]. Local governments may need to put contingency plans in place for those at high risk of complications related to food supply instability. The presence of certain contaminants in water can lead to health issues, including gastrointestinal illness, reproductive problems, and neurological disorders [61]. State and local health or environmental departments often test for nitrates, total coliforms, fecal coliforms, volatile compounds, and pH. Health or environmental departments in city or county governments

are suggested to maintain a list of state-certified (licensed) laboratories that test for a variety of Water Quality Indicators (WQIs) and contaminants [61].

### 3.10. Water and Sewer

This domain of action interacts with the climate change areas that correspond to human interaction with water: *extreme weather*, *precipitation*, *sea level and coastal areas*, and *water quality and supply*.

Extreme weather, rising sea levels, and storms will result in new challenges for water and sewer systems. Studies show that extreme precipitation events increase the level of contaminants in waterways and increase the risk of illness [62,63]. Existing wastewater and sewage infrastructure will likely require adaptive upgrades to accommodate for these events and improve surveillance and identification of disease vectors in the treated water [64]. The overflow of sewage and storm water treatment plans, caused by flooding events, can also quickly spread disease associated with E. coli and other bacteria [46]. Operations systems of coastal municipalities will be increasingly compromised by rising sea levels, requiring them to invest in new methods and technology like enhanced tide-gate programs and pumps capable of overcoming back surge problems. Some cities may need to expand their overall pumping capacities, which are expensive and energy intensive [53]. Furthermore, existing deficiencies in infrastructure and storm drainage systems must be remedied to mitigate the rising risk of water-borne bacterial contamination in public water [64].

## 4. Discussion

Our preliminary analysis of the case of urban climate policy reveals useful information for both scientists and policymakers. The blank spots on the map indicate where there is an unmet need for policy-relevant science, and therefore areas that scientists should focus on if they are interested in producing knowledge that helps cities adapt to climate change [23]. For example, there is relatively little policy-relevant work done in the category of *ecosystems and biodiversity*. A biodiversity scientist looking to contribute to urban sustainability could focus on highlighting, say, how infrastructure and urban planning interacts with ecosystems and biodiversity, and be relatively confident that the research would be filling a need. Alternatively, a city that is interested in making science-informed policy could commission a study in this area with confidence that the relevant scientific expertise could be found. These are fertile sites for the co-production of knowledge.

The logic synchronization framework also reveals the areas where there is significant policy-relevant science already available. These are policy domains that benefit from of large pools of relevant research, and therefore are potential areas of focus for those looking to create science-informed policy. For example, since there is significant literature at the intersection of precipitation and emergency management, creating science-informed policy in this area should be relatively easy.

The logics framework also helps us make sense of other attempts to bridge the gap between science and policy. The 'policy scientist' framework proposed by Harold Lasswell and Myres McDougal, for example, imagines scientists as practitioners "of a kind of science that . . . [puts] the methods and findings of a general science to work in solving real-world problems" [65,66]. This influential body of thought, however, is "torn between positivist canons of "pure science" and faith in the discipline to inform . . . decision making;" or, in other words, between the logic of inquiry and the logic of action [66]. The 'policy scientist,' along with the scientist working on actionable science can find it difficult to satisfy the dictates of both the logic of inquiry and the logic of action. This is the tension that makes the work so difficult.

Boundary organizations are another way people have attempted to connect science and policy. These organizations exist 'between' scientific and policy domains and are accountable to both communities. This type of organization "facilitates collaboration between scientists and nonscientists, and they create the combined scientific and social

order" by producing artifacts and knowledge which are intelligible to both scientific and policy communities [67]. This requires that boundary organizations understand both the logic of inquiry and the logic of action, and either 'code switch' between them or include elements of each in their work. These organizations often orient their work to the areas where the logics are aligned and both communities can be satisfied; they therefore employ a version of the logic synchronization method similar to what we discuss here.

With regards to urban climate policy, boundary organizations are recognized as having an important and effective role as a conduit of knowledge from climate scientists to policymakers [8,16,24]. Examples of boundary organizations in this area are the International Council for Local Environmental Initiatives (ICLEI) and the Urban Climate Change Research Network (UCCR) [68,69]. Yet even these organizations reveal an inclination to favor one logic over the other. UCCR is composed of "scholars and experts from universities and research organizations," [69]—which implies that they operate under the logic of inquiry—while the ICLEI is a "global network of more than 1750 local and regional governments" [70]—which implies a deference to the logic of action. Although both straddle the boundary between science and policy, they have their roots in different logics.

Even in the case of boundary organizations, the logics remain incompletely synchronized. The UCCR, the ICLEI, and other boundary organizations would be ideal users of our tool, as they are positioned to do a much more thorough analysis than the one presented here, and such an analysis would enhance their understanding of the boundary between science and policy that they are attempting to bridge. A better understanding of how and where the logics of inquiry and action synchronize would enhance their effectiveness as boundary organizations.

## 5. Conclusions

It is sometimes assumed that scientific knowledge, once produced and released to the world, will be taken up by policymakers simply because it is relevant [71]. Yet the production of knowledge is guided by a different logic than its consumption, and producing knowledge does not guarantee its use [7]. Science has a reputation for bringing too much of the wrong type of information to the table [4]. Its operational logic (the logic of inquiry) leads to places that policymakers (who are guided by the logic of action) simply don't need to go. Even though scientists may produce knowledge which is 'actionable,' it remains true that taking action does not require scientific knowledge. Policymaking advances regardless of what science is available.

Yet scientists and policymakers often have the same broad goals (such as the establishment of effective urban climate policy) and both groups recognize the importance of working with the other [22,23,72]. They should—and can—find ways to work together. Knowledge co-production and actionable science offer promising models for how to approach this kind of cross-sector work [14,16,21]. Yet even these models can produce knowledge for which there is weak demand, while overlooking areas where demand is strong. Our work complements this body of thought by supplying a theoretical framework and a basic tool which allows researchers and policymakers to identify areas of fruitful and smooth collaboration and coproduction, where the logics of inquiry and action are synchronized.

In this paper we introduced the logics of action and inquiry, and the corresponding concept of logic synchronization. We showed using the case of urban climate change policy how these concepts could be used to map and explain the disconnect between scientific research and policy action, even in cases where researchers strive to make their knowledge actionable.

We believe that the dynamics discussed in this paper, between the logics of inquiry and action, are universal. The logic synchronization concept can be applied to the intersection of any subject area and any domain of action. The validity of the results, however, will depend on the rigor of the analysis, which in turn likely depends on the researcher's familiarity with the subject area. Our analysis of urban climate science is limited by our

lack of expertise in this area and our use of a purposive literature review; therefore our results with regard to the specific domain of urban climate policy are only tentative. As policy scholars, we are not as well equipped for this analysis as an expert in urban climate science would be. We encourage subject-area experts to pursue a more rigorous analysis than we presented here. We also encourage experts in other areas to do similar analyses with other subject areas and other institutions (including other levels of government). An analysis such as ours, which maps the areas of action for a specific group or institution to areas of inquiry in a specific scientific domain, can reveal both areas of opportunity for science-informed action and where further research will need to be done.

This paper is exploratory rather than definitive; further research is needed to understand how the dynamics we have identified here interact with other factors and pressures in the process of policymaking. Further research could also explore how the dynamics of logic synchronization play out differently in countries with different approaches to science and government.

Learning to identify where and how the logics synchronize will be important for both policymakers and scientists if they hope to accomplish their shared goal of creating science-informed policy. The approach we demonstrated in this paper can reveal the topology of an area defined by the intersection of a specific level of policymaking and a specific domain of scientific research. This allows the scientists and policymakers involved to reach a common understanding of the dynamics of their field and to identify areas of opportunity for science-informed policy, policy-relevant science, and deeper collaboration.

**Author Contributions:** Conceptualization, D.M.A. and C.J.B.; methodology, D.M.A. and C.J.B.; data curation, Q.W and D.A.C.; writing—original draft preparation, Q.W. and D.A.C.; writing—review and editing, C.J.B.; visualization, Q.W., D.A.C. and C.J.B.; supervision, D.M.A.; project administration, D.M.A.; funding acquisition, D.M.A. All authors have read and agreed to the published version of the manuscript.

**Funding:** This research received no external funding.

**Acknowledgments:** The authors would like to thank and acknowledge Elizabeth Kuttner, Reyna Olvey, Vincent Kao, Katje Benoit and Melody Moss, who helped with background research and editing in early versions of this project.

**Conflicts of Interest:** The authors declare no conflict of interest.

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
