# Peer review of "Synchronizing the Logic of Inquiry with the Logic of Action: The Case of Urban Climate Policy"

_sustainability, doi:10.3390/su131910625_

Round 1

Reviewer 1 Report

Synchronizing the logic of scientific inquiry with the logic of policy action: the case of urban climate policy

The topic of this research is interesting and indeed critical to the respective contributions of researchers and policy makers. Even more so in the field of the fight against climate change. The study is interesting but there is a significant need for clarification of the methodology. This is the main aspect at this stage, and it is specified below throughout the text and in a general way.

L184-186: “policy-relevant research relating to the effects of climate change on cities was collected and organized according to dominant themes.

Where were they collected? How? By keyword? Over what period? for what geographical area? The same as for the policy action scope?

The authors should clarify the methodology for determining the categories.

In the same way, as regards the scope of possible actions, was the review done over a period of time? At a particular date? Consistent with the definition of the scopes of policy-relevant research?

L188-189 […] L219-222: “Based on their organizational structure (i.e. named municipal departments), we identified ten broad areas of action” [….] “We then recorded where we were able to locate interaction between these domains in the literature.”

Has this association been made only with regard to the names of the departments and not with regard to the actions actually carried out?

The policy actions are very general and broad.

Even if some precisions are brought later in the paper to better define these limits, this should be specified more directly as to the method followed to collect and synchronize.

L233-234: “for example, one might record the number of journal articles in each area, but weight each article based on its relevance to a particular policy domain”

How would you determine the weights? Its relevance?

L235-236: “We want to be clear that these insights are based only on preliminary assessment of a purposive,”

How was the sample determined?

Even if “Our intention is primarily to demonstrate our tool”, this is important information

The authors should still give a brief indication of what the field of urban climate science encompasses, in addition to the reference to Rosenzweig et al. , so that we can already have an idea of the scope and purpose

L244-245: “Below we analyze the literature on climate change which relates to each domain of government action. Each section corresponds to a row on the table above”

Can the authors elaborate on how they tried to determine this and how many linked items it represents?

Should they be considered on the basis of the citations made in the paper for each category?

Were there other papers considered that were not relevant to the classification?

The authors do not give much indication of the degree or level of relationship

On several occasions, the authors refer to urban climate policy issues in the context of developing countries. If the geographical area covered by the scientific papers is indeed global and therefore includes developing and developed countries, is the classification of the logic of actions based on the 10 largest American cities relevant? The doubt is allowed at this level or would deserve justification.

The authors show well the connections (or disconnections) between the research topics for the logic of inquiry, and the themes of the logic of actions, but only the conjunction of the themes and not the actual actions behind them, or even, if there are indeed actions behind it and how they overlap. I have difficulty understanding the passage at this stage, the logic of synchronization or how actions (or the process of policymaking) are decoupled from science, to see what guarantees the use of science in the definition of actions. Or the purpose was to identify the blank spots on the map?

Author Response

Thank you for reviewing our paper. The comments we received were useful, both for helping us clarify the paper and revealing a way we may have confused our readers.  

Many of the comments from the reviewers are about how we conducted our literature review. We have added more information about our process, but we want to be clear that this paper is a theoretical discussion that uses urban climate policy as an example to help illustrate our point. We have no hypothesis, data, or findings, and therefore ‘methods’ and ‘results’ may be a misleading way to frame the sections of our paper. We have therefore renamed section two “An Approach to Logic Synchronization” (instead of “Materials and Methods”) and section three “Making sense of Interacting Logics” (instead of “Results”). We hope this makes the paper clearer for future readers.

All authors have participated in this revision. The ten-day turnaround time limited how deeply we could revise the paper; we believe that we have responded to the issues raised by the reviewers, however, we would be happy to make further changes should they be necessary. 

Reviewer 1:

Comment:
The topic of this research is interesting and indeed critical to the respective contributions of researchers and policy makers. Even more so in the field of the fight against climate change. The study is interesting but there is a significant need for clarification of the methodology. This is the main aspect at this stage, and it is specified below throughout the text and in a general way.

L184-186: “policy-relevant research relating to the effects of climate change on cities was collected and organized according to dominant themes.”

Where were they collected? How? By keyword? Over what period? for what geographical area? The same as for the policy action scope?

Response:
We are glad that the reviewer finds our work interesting – and critical, no less.

As noted above, this paper does not have ‘methods’ in the traditional sense, and so the task of clarifying our methods is a tricky one. We have, however, attempted to clarify how we found the literature we used to demonstrate the logic synchronization concept.

Reviewed literature was collected via an online search of peer-reviewed journals and grey literature. This review was conducted in 2017, with the earliest notable literature coming from the early 1990s. There was not a specific focus on geographic area, since the effects of climate change are global in nature; however, the review focused on English-language literature.

If you believe it would be useful, we can also include information on what search terms we used to find this literature:

Search terms used included: 

  • Climate change impact on
    • Cities
    • Urban 
  • Climate change impact on: 
    • Public health 
    • Human health 
    • Biodiversity 
    • The environment 
    • Government 
    • Public Policy 
    • Public Management 
    • Education 
    • Quality of life 
    • Infrastructure 
    • Water resources 
    • Economics 
    • Economic development 
    • Urban development 
    • Energy 
    • Natural disasters 
    • Coastal populations 
    • Agriculture 
    • Farming 
    • Technology 

Comment:

The authors should clarify the methodology for determining the categories.

In the same way, as regards the scope of possible actions, was the review done over a period of time? At a particular date? Consistent with the definition of the scopes of policy-relevant research?

Response:

We established the policy action scope by searching on city government websites in 2017. To determine the domain of action, we reviewed the structure of the 10 largest cities in the US (New York, Los Angeles, Chicago, Houston, Phoenix, Philadelphia, San Antonio, San Diego, Dallas, and San Jose) and 10 largest cities in China (Shanghai, Beijing, Tianjin, Shenzhen, Guangzhou, Chengdu, Chongqing, Dongguan, Shenyang, Wuhan) to establish the scope of policy action.

The fact that we used Chinese cities as well as American ones was not mentioned in our original submission. This fact was uncovered when we returned to the original documents used during the first stages of this project, I order to clarify our process. We appreciate that the reviewer’s question prompted us to revisit these documents, as the addition of Chinese cities expands the applicability of our paper.

As we stated above, we searched the scopes of possible actions during 2017, as that was when work on this paper began.

Comment

L188-189 […] L219-222: “Based on their organizational structure (i.e. named municipal departments), we identified ten broad areas of action” [….] “We then recorded where we were able to locate interaction between these domains in the literature.”

Has this association been made only with regard to the names of the departments and not with regard to the actions actually carried out?

Response:

We recognize that the actions a city government might take do not always map cleanly onto the city’s organizational structure. However, we also believe that a city’s organizational structure constrains and guides the actions which the city can take. Although we did pay attention to the actual actions that municipal departments perform, we largely chose to accept the city governments as they presented themselves, organized by departments.

We fully acknowledge that there are many ways to categorize ‘areas of action;’ however, the underlying dynamics of logic synchronization remain the same. 

Comment:

The policy actions are very general and broad.

Even if some precisions are brought later in the paper to better define these limits, this should be specified more directly as to the method followed to collect and synchronize.

Response:

Since we hope to provide a general framework for synchronization of inquiry and action, we consider the broadness of the identified areas of policy action to be an asset.

Comment:

L233-234: “for example, one might record the number of journal articles in each area, but weight each article based on its relevance to a particular policy domain”

How would you determine the weights? Its relevance?

Response:

This would depend on the goals of the research being conducted. For example, if one were looking at misinformation on social media one might weigh independent research more heavily than research conducted within social media companies. Or if one was researching blockchain’s use in humanitarian situations, one might weigh research by development scholars more than research by those who study the technical aspects of blockchain.

This is a good question which studies which build on ours must consider. However, since the answer to this question relies on the specifics of the research being conducted, we believe it would be confusing to address this question specifically in the paper.

Comment:

L235-236: “We want to be clear that these insights are based only on preliminary assessment of a purposive sample,”

How was the sample determined?

Even if “Our intention is primarily to demonstrate our tool”, this is important information

 Response:

See above for more information about how we found the literature on urban climate change used for this paper.

Comment:

The authors should still give a brief indication of what the field of urban climate science encompasses, in addition to the reference to Rosenzweig et al., so that we can already have an idea of the scope and purpose.

Response:

We were not sure what the reviewer means by “what the field encompasses,” however we interpreted it as a request to clarify the disciplinary domain. Therefore, the following line has been added:
“Urban climate science is a dynamic, interdisciplinary field which brings together climate scientists, social scientists, policy scholars, and urban planners.”

Comment:

L244-245: “Below we analyze the literature on climate change which relates to each domain of government action. Each section corresponds to a row on the table above”

Can the authors elaborate on how they tried to determine this and how many linked items it represents?

Should they be considered on the basis of the citations made in the paper for each category?

Were there other papers considered that were not relevant to the classification?

The authors do not give much indication of the degree or level of relationship

Response:

The following sentence has been added to the paper, following the section quoted in the comment:

“We explore where the logics of inquiry and action seem to be synchronized, as indicated by the availability of literature which speaks to both a domain of action and a domain of inquiry.”

We also state on line 235 that “We chose to use a binary scale (synchronized / not synchronized) to represent where we identified logic synchronization, in the form of policy-relevant science. Although not all areas had comparable levels of interaction (which can be seen in the discussion section below), our review of this literature was not systematic and thus we are unable to make claims about the relative volume of work done in each of these areas.”

Comment:

On several occasions, the authors refer to urban climate policy issues in the context of developing countries. If the geographical area covered by the scientific papers is indeed global and therefore includes developing and developed countries, is the classification of the logic of actions based on the 10 largest American cities relevant? The doubt is allowed at this level or would deserve justification.

Response:

This was a very useful comment, and we wish to thank the reviewer for pointing this out.

This comment caused us to revisit the very first drafts of this paper. It turns out that the initial classification of logic was based on not just American cities, but also Chinese cities. This fact had gotten lost between drafts but has now been reincorporated into the paper.

Line 193 now states:

“To establish the scope of policy action, we reviewed the structure of the 10 largest cities in the US (New York, Los Angeles, Chicago, Houston, Phoenix, Philadelphia, San Antonio, San Diego, Dallas, and San Jose), as well as the 10 largest cities in China (Shanghai, Beijing, Tianjin, Shenzhen, Guangzhou, Chengdu, Chongqing, Dongguan, Shenyang, Wuhan). The US and China were chosen because they are the world’s largest and most influential developed and developing countries.”

Comment:

The authors show well the connections (or disconnections) between the research topics for the logic of inquiry, and the themes of the logic of actions, but only the conjunction of the themes and not the actual actions behind them, or even, if there are indeed actions behind it and how they overlap. I have difficulty understanding the passage at this stage, the logic of synchronization, or how actions (or the process of policymaking) are decoupled from science, to see what guarantees the use of science in the definition of actions. Or the purpose was to identify the blank spots on the map?

Response:

The reviewer is right that the purpose of the exercise as presented here is to identify the blank spots on the map – as well as the spots which are not blank. As we discuss around line 486, blank spots indicate areas where scientists can focus on creating new knowledge, confident that they will find interested policymakers. On the other hand, spots which are NOT blank indicate to policymakers areas where there exists science upon which they can build policy.

Whether this policy translates into actions depends on a number of other factors which are beyond the scope of this paper: the effectiveness of the municipal government, the responsiveness of the urban population, the presence or absence of political will necessary to implement these policies, etc.

The question of guaranteeing the use of science in action is several steps removed from what is discussed in this paper. Even when the logic of inquiry and the logic of action is synchronized and science-informed policy is created, science-informed action may be impeded for other reasons. However, where the logic is NOT synchronized, the science-informed policy is impossible. 

Please find the revised manuscript in the attachment.

Reviewer 2 Report

In my opinion, this is a missing piece of research. However, it would be important to state in the title that the research was carried out in the United States and that the findings are primarily valid in this context. The issues of extension should be examined in more detail in the discussion chapter. 
Although the article is based on a large literature review, it would be important to refer to more recent articles published in 2020-21.

Author Response

Reviewer #2

Comment:

In my opinion, this is a missing piece of research. However, it would be important to state in the title that the research was carried out in the United States and that the findings are primarily valid in this context. The issues of extension should be examined in more detail in the discussion chapter.

Although the article is based on a large literature review, it would be important to refer to more recent articles published in 2020-21.

Response:

We are glad that the reviewer finds out research of value.

The domains of action discussed in this paper derive from an analysis of both American cities and Chinese cities. The domains of inquiry discussed here derive from an analysis of the English language literature around urban climate change, however there is no reason to believe that the dynamics of climate change are different in countries which do not speak English. In fact, much of the research cited is not based in the US.

Furthermore, the dynamics between the logic of inquiry and the logic of action – which are the main topic of this paper - are not specific to the US. We have added to the discussion section at line 509:
“Further research could explore how the dynamics of logic synchronization play out differently in countries with different approaches to science and government.”

The short turnaround time for this review made it impossible for us to fully analyze more recent publications against our framework. If the editor is willing to extend the revision deadline, we will happily incorporate literature from 2020 and 2021.

Please find the revised manuscript in the attachment.

Round 2

Reviewer 1 Report

I thank the authors for all these precisions. 

I believe it would be useful, they include more information on how they found their literature (what search terms they use, where they searched, on what period) in the paper (even in footnote if they prefer).  It is useful to perceive the perimeter and the possible limits for future explorations

I am quite surprised that they did not mention from the beginning the reference to the 10 Chinese cities, it is an important information for the analysis I think.

I am not completely convinced by the method but the questioning and the analysis are interesting 

Author Response

Comment:
I thank the authors for all these precisions.

Response:
We thank the reviewer for their comments. They have helped make this a much better paper.

Comment:
I believe it would be useful, they include more information on how they found their literature (what search terms they use, where they searched, on what period) in the paper (even in footnote if they prefer).  It is useful to perceive the perimeter and the possible limits for future explorations

Response:
We have taken the reviewer’s suggestion and added a footnote in Line 188 of the manuscript explaining more of how we found the literature used in this paper. Please see the footnote in the manuscript.

We could not find information on how the journal wants footnotes to be formatted, so if it needs to be adjusted we are happy to make the change.

Comment:
I am quite surprised that they did not mention from the beginning the reference to the 10 Chinese cities, it is an important information for the analysis I think.

Response:  This is a lesson for us in the perils of working with large teams over long periods of time – there are opportunities for information loss.

We appreciate that the reviewer’s questions gave us reason to revisit the early documentation on this project. We were able to both improve the paper and reflect on how we can improve our work processes.

Comment:
I am not completely convinced by the method but the questioning and the analysis are interesting

Response:
We hope that the addition of the footnote further clarifies how we found the literature we used in this paper. Ours is a theoretical paper introducing a concept, and so we would like to emphasize that we do not have a ‘method’ like one would find in an empirical paper that explores a hypothesis. We recognize that the way the paper was initially structured was misleading, and we hope that the current version fixes this issue.

We are glad that the reviewer finds the questioning and analysis interesting. We believe that this paper’s contribution really does lie in these areas – the section on urban climate change simply demonstrates the ideas we present in the other sections.